

# Neural noiseprint transfer: a generic noiseprint-based counter forensics framework

Ahmed Elliethy

Department of Electrical and Computer Engineering, Military Technical College, Cairo, Egypt

## ABSTRACT

A noiseprint is a camera-related artifact that can be extracted from an image to serve as a powerful tool for several forensic tasks. The noiseprint is built with a deep learning data-driven approach that is trained to produce unique noise residuals with clear traces of camera-related artifacts. This data-driven approach results in a complex relationship that governs the noiseprint with the input image, making it challenging to attack. This article proposes a novel neural noiseprint transfer framework for noiseprint-based counter forensics. Given an authentic image and a forged image, the proposed framework synthesizes a newly generated image that is visually imperceptible to the forged image, but its noiseprint is very close to the noiseprint of the authentic one, to make it appear as if it is authentic and thus renders the noiseprint-based forensics ineffective. Based on deep content and noiseprint representations of the forged and authentic images, we implement the proposed framework in two different approaches. The first is an optimization-based approach that synthesizes the generated image by minimizing the difference between its content representation with the content representation of the forged image while, at the same time, minimizing the noiseprint representation difference from the authentic one. The second approach is a noiseprint injection-based approach, which first trains a novel neural noiseprint-injector network that can inject the noiseprint of an image into another one. Then, the trained noiseprint-injector is used to inject the noiseprint from the authentic image into the forged one to produce the generated image. The proposed approaches are generic and do not require training for specific images or camera models. Both approaches are evaluated on several datasets against two common forensic tasks: the forgery localization and camera source identification tasks. In the two tasks, the proposed approaches are able to significantly reduce several forensic accuracy scores compared with two noiseprint-based forensics methods while at the same time producing high-fidelity images. On the DSO-1 dataset, the reduction in the forensic accuracy scores has an average of 75%, while the produced images have an average PSNR of 31.5 dB and SSIM of 0.9. The source code of the proposed approaches is available on GitHub (https://github.com/ahmed-elliethy/nnt).

Corresponding author
Ahmed Elliethy,
a.s.elliethy@mtc.edu.eg

# INTRODUCTION

Digital image forensics is a broad field concerned with applying imaging science and domain expertise to provide legal aspects for the content of a digital image. Digital image forensics attracts attention nowadays because of the increasing popularity of digital cameras and the availability and ease of use of powerful image editing software. Forensic analysts can employ traces of an image to detect complex forgeries, localize falsified content, and attribute the image to the camera that acquires it with great accuracy (*Ferreira et al., 2020*).

There exist a vast amount of different forensic methods. One of the most successful methods is the method that utilizes imperfections in manufacturing the sensing elements of a digital camera. These imperfections make the different sensing elements produce different values when exposed to the same light intensity. Such variations of light sensitivity are called photo response non-uniformity (PRNU). The PRNU pattern is unique for each camera, and thus, it can be regarded as a fingerprint for the camera (*Chen et al., 2008*). This makes the PRNU pattern serves as a powerful tool for several forensic tasks such as camera source identification, integrity verification, and authentication (*Cao & Kot, 2012*; *Conotter, Comesana & Perez-Gonzalez, 2015*). Additionally, there exists a well-posed mathematical relation relating an acquired image from a camera with its PRNU (*Fridrich, 2009*). Specifically, the output of a digital camera's sensor is mathematically related to the incident light intensity at each pixel and the PRNU (see Eq. (1) in *Fridrich (2009)*). However, this mathematical relation makes PRNU-based forensics vulnerable to several attacks (*Picetti et al., 2022*; *Raj & Sankar, 2019*; *Elliethy & Sharma, 2016*; *García Villalba et al., 2017*).

With the recent success of deep learning in several tasks, a new data-driven approach is proposed (*Cozzolino & Verdoliva, 2020*) to extract camera-related artifacts from an image. These artifacts are named as *noiseprint*. The noiseprint serves as a fingerprint of a camera, much like the PRNU. However, the extraction of the noiseprint is performed differently. Specifically, twin convolutional neural networks (CNNs) are organized into a Siamese architecture and fed several image patches drawn from two classes. The positive class represents patches cropped from images at the same sensor spatial location and acquired by the same camera. The other is the negative class for other patches obtained from different cameras or locations. The twin CNNs are trained to produce the same output for patches from the positive class while producing different outputs for negative patches. After the training is performed, the weights of each CNN are frozen, and the CNN produces the noiseprint for a given image that contains clear traces of unique camera artifacts. This makes the noiseprint an ideal tool for several forensic applications as proposed in *Cozzolino Giovanni Poggi Luisa Verdoliva (2019)*, *Marra et al. (2020)* and *Cozzolino et al. (2020)*. Additionally, the noiseprint becomes an essential building block in many advanced recent forensics algorithms (*Mareen et al., 2022*; *Marra et al., 2020*) that blend the noiseprint with other fingerprints (or features from other domains) to achieve superior performance in different forensics tasks. Thus, we can treat the noiseprint as a cornerstone in a family of related algorithms that handle different forensics tasks. Therefore, proposing a counter-forensics framework for attacking the noiseprint is very important in

digital forensics in general. However, the complex training of the twin CNN to produce the noiseprint results in a complex relationship that governs the noiseprint with the input image. Therefore, attacking the noiseprint-based forensics is challenging.

Recently, generative adversarial networks (GANs) have been successfully applied in several counter forensic tasks (*Stamm & Zhao, 2022*). In *Chen, Zhao & Stamm (2019)*, a GAN is employed to perform camera model attacks on a specific camera model. Specifically, a generative network is trained to generate an image with falsified camera model traces, while the discriminator network is trained to distinguish the generated images from the real ones. Another counter-forensic method is proposed in *Güera et al. (2017)* that employs an adversarial image generator that takes an input image and introduces subtle perturbations to misclassify the estimated camera model when analyzed by a CNN-based camera model detector. In *Wu & Sun (2021)*, a new GAN-based approach is proposed to deal with traces left by multiple manipulating operations. In *Cozzolino et al. (2021)*, a GAN is trained to generate a synthetic image that is visually similar to an input image and inserts a target camera's traces in the generated synthetic image. The generated synthetic images deceive several CNN-based camera model identification detectors, making them believe that the target camera model took the images. Despite the effectiveness of the GAN-based methods in fooling the camera model detectors, training of the GAN is required for each camera model. In other words, for each camera model, a different generator-discriminator couple must be trained to generate images from the camera model. Additionally, these methods require a training dataset containing authentic images of the camera model. This makes these methods less general and limits their applicability in anti-forensic tasks.

To the best of our knowledge, there is no generic approach for attacking the noiseprint-based forensic methods. In this article, we take the first step in this direction and propose a novel generic framework for attacking the noiseprint-based forensic methods. Specifically, the proposed framework successfully synthesizes a new image that is visually similar to a forged one but simultaneously transfers the noiseprint from an authentic image into the synthesized image to make it appear as if it is authentic. To do so, we propose two approaches within this framework. The first is an optimization-based approach that synthesizes a generated image by minimizing the difference between its content representation with the content representation of the forged image while, at the same time, minimizing the noiseprint representation difference with the authentic one. The second approach is a noiseprint injection-based approach, which first trains a novel neural noiseprint-injector network that can inject the noiseprint of an image into another one. Then, the trained noiseprint-injector is used to inject the noiseprint from any authentic image into the forged image to produce the generated image. The proposed approaches are generic and do not require training for specific images or camera models. Additionally, we evaluated the effectiveness of the proposed approaches against two common forensic tasks, which are the forgery localization and camera source identification tasks. The proposed approaches are able to significantly reduce several forensic accuracy scores compared with two noiseprint-based forensics methods (*Cozzolino & Verdoliva, 2020*; *Mareen et al., 2022*) while at the same time producing high-fidelity images.

In summary, the contributions of this article are:

- To the best of our knowledge, we propose the first generic framework for attacking the noiseprint-based forensic methods. The proposed framework is general and does not require training for specific images or camera models.
- Within the proposed framework, we propose two approaches for attacking the noiseprint-based forensic methods. One formulates and solves an optimization problem, while the other is based on training a novel neural noiseprint-injector network.
- We present an extensive experimental analysis that evaluates the proposed approaches in attacking two common forensic tasks: the forgery localization and camera source identification tasks.

The remainder of this article is organized as follows. 'Noiseprint-based forensics' presents an overview of the noiseprint-based forensic method (*Cozzolino & Verdoliva, 2020*). 'Deep feature representations and proposed neural noiseprint transfer framework' presents the proposed counter-forensic approaches. 'Experimental analysis' presents our experimental analysis that evaluates the performance of the proposed approaches on two common counter-forensic tasks. Finally, we summarize our conclusion in the 'Conclusion'.

## NOISEPRINT-BASED FORENSICS

This section gives an overview of the noiseprint-based forensic method proposed in *Cozzolino & Verdoliva (2020)*. As shown in Fig. 1, the method can identify and localize a forgery in a digital image. Specifically, the method first extracts the so-called noiseprint from the image. Then the method localizes the forgery by searching for inconsistencies in the noiseprint using the blind localization algorithm proposed in *Cozzolino, Poggi & Verdoliva (2015)*. As shown in the figure, the noiseprint shows inconsistencies at the most right, as the boat shown there is copied from another image, *i.e.,* forged.

The noiseprint extraction method in *Cozzolino & Verdoliva (2020)* employs a CNN-based image de-noising approach (*Zhang et al., 2017*) that is originally trained to extract noise residual from an image. However, the noise residual obtained by the denoiser (*Zhang et al., 2017*) is suitable for de-noising applications and can not be directly utilized for forensic purposes. Therefore, the noiseprint extraction method (*Cozzolino & Verdoliva, 2020*) trains the denoiser differently so that the output noiseprint differentiates between the authentic and forged images, as shown in Fig. 1. This training methodology is depicted in Fig. 2.

Specifically, twin CNN denoisers are organized into a Siamese architecture and fed several image patches drawn from either positive or negative classes. The positive class represents the patches from the same camera at the exact sensor-level spatial location. In contrast, the negative class represents other patches obtained at different locations or from different camera models. The output of the twin denoisers is presented to a binary classifier with cross-entropy loss, as shown in Fig. 2. The twin denoisers are trained to minimize the loss. This encourages the CNN denoiser to generate a similar output for positive class patches and a different output for negative ones. This modification in the training

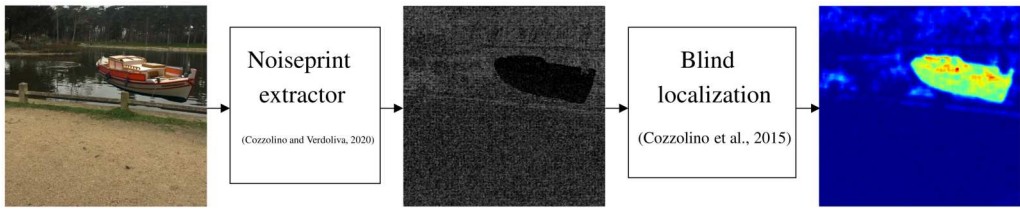

**Figure 1** Noiseprint-based image forensic.

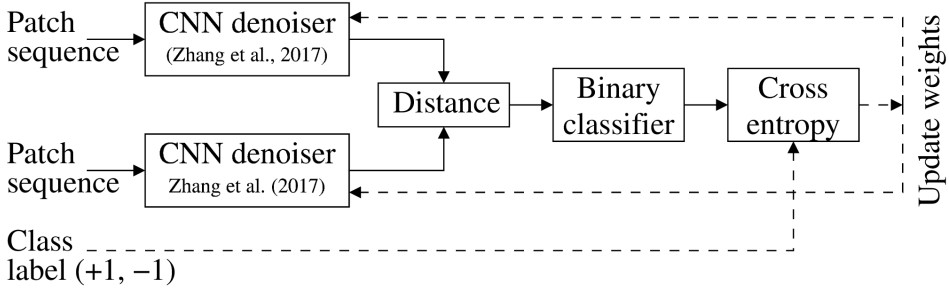

**Figure 2** Noiseprint training methodology (*Cozzolino & Verdoliva, 2020*). Twin CNN denoisers are organized into a Siamese architecture and trained to minimize the cross-entropy loss of a binary classifier. This encourages the denoisers to generate a similar output for positive class patches (obtained from the same camera at the same sensor-level spatial location) and a different output for the negative class patches.

methodology allows the CNN denoiser (*Zhang et al., 2017*) to output the noiseprint, instead of the original noise residual, to be suitable for forensic applications.

As discussed above, the noiseprint extraction is performed based on complex training for a CNN denoiser. The extracted noiseprint has a complex mathematical relation that governs its relationship with the input image. Thus, the noiseprint can not be spoofed by a simple inversion process or manipulation, as in the case of PRNU.

The following section presents the proposed neural noiseprint transfer framework for noiseprint counter forensics. With a given forged and authentic image pair, the proposed framework successfully transfers the noiseprint from the authentic image to a newly generated image that is imperceptible to the forged image. However, its noiseprint looks very typical, as if it was obtained authentically. Thus, the proposed framework renders the noiseprint-based forensics ineffective.

## DEEP FEATURE REPRESENTATIONS AND PROPOSED NEURAL NOISEPRINT TRANSFER FRAMEWORK

In this section, we present the neural noiseprint transfer framework. As shown in Fig. 3, given an authentic image $I_a$ and a forged image $I_f$, the goal of the proposed framework is to generate a new image $I_g$ that is visually imperceptible to $I_f$, but its noiseprint is very close to the noiseprint of $I_a$. To do so, we propose two approaches. The first approach is

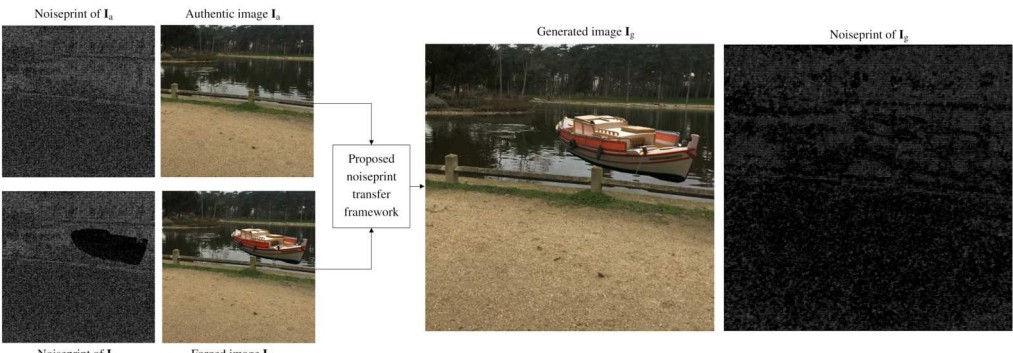

Noiseprint of $\mathbf{I}_a$   Authentic image $\mathbf{I}_a$   Generated image $\mathbf{I}_g$   Noiseprint of $\mathbf{I}_g$

Proposed noiseprint transfer framework

Noiseprint of $\mathbf{I}_f$   Forged image $\mathbf{I}_f$

**Figure 3** **Block diagram of the proposed noiseprint counter forensics.** Given an authentic image $\mathbf{I}_a$ and a forged image $\mathbf{I}_f$, the goal of the proposed approaches is to generate a new image $\mathbf{I}_g$ that is visually imperceptible to $\mathbf{I}_f$, but its noiseprint is very close to the noiseprint of $\mathbf{I}_a$.

an *optimization* based approach that estimates $\mathbf{I}_g$ by minimizing the difference between its content representation with the content representation of $\mathbf{I}_f$, while at the same time, minimizing the noiseprint representation difference with $\mathbf{I}_a$. The second approach is a *noiseprint-injection* based approach, which first trains a novel neural noiseprint-injector network that can inject the noiseprint of an image into another. Then, the trained noiseprint-injector is used to inject the noiseprint from $\mathbf{I}_a$ into $\mathbf{I}_f$ to produce $\mathbf{I}_g$.

To define suitable content and noiseprint representations of an image to be used in the proposed approaches, we first present a critical observation regarding the visualization of the deep feature representations of the CNN denoiser (*Zhang et al., 2017*) that was used for the noiseprint extraction, as we discussed in 'Noiseprint-based forensics'. Then, based on this observation, we present a definition for the image's content and noiseprint representations and employ the definitions in the proposed approaches we present afterwords.

## Deep feature representations

We studied the deep feature representations of the CNN denoiser (*Zhang et al., 2017*) in a similar way that was used before toward the goal of understanding the deep image representations (*Mahendran & Vedaldi, 2015*; *Lin & Maji, 2016*) and for artistic style transfer approaches (*Gatys, Ecker & Bethge, 2016*; *Johnson, Alahi & Fei-Fei, 2016*). Specifically, we present an input image $\mathbf{I}$ to the CNN denoiser and produce the features at a specific convolutional layer $l$. Then, we find another image $\mathbf{J}$ that its features at the same convolutional layer best match the features produced for $\mathbf{I}$. Mathematically,

$$\mathbf{J} = \arg \min_{\mathbf{X}} \|\mathcal{F}^l(\mathbf{I}) - \mathcal{F}^l(\mathbf{X})\|_2^2, \tag{1}$$

where $\mathcal{F}^l(\mathbf{X})$ is the features produced for image $\mathbf{X}$ at the convolutional layer $l$.

Figure 4 shows the produced images obtained from the features of different convolutional layers of the CNN denoiser using Eq. (1). As shown in the figure, the images produced from the features of the early layers tend to preserve the content of the input image. However, as we go deeper in the network, the produced images tend to lose this content fidelity

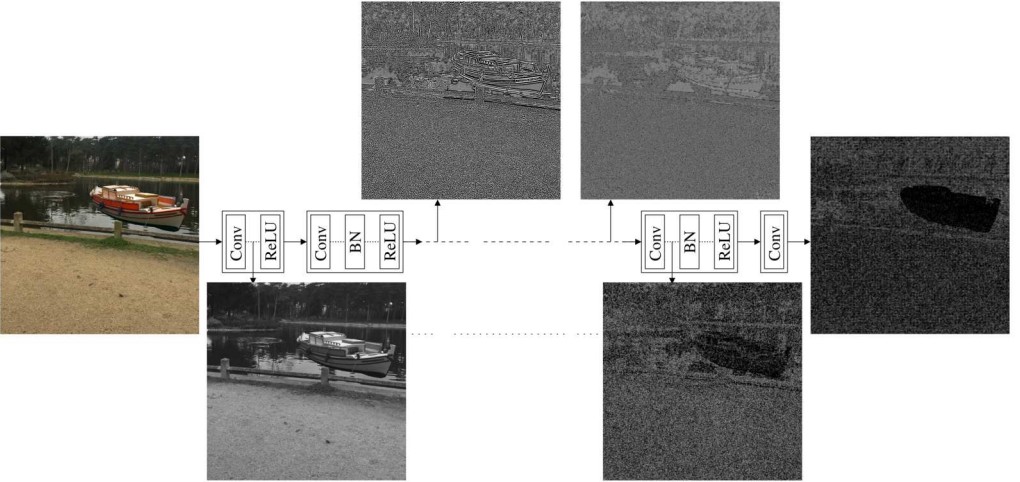

**Figure 4** **Architecture of the CNN denoiser (*Zhang et al., 2017*) that was used for the noiseprint extraction.** The architecture comprises several convolutional(Conv), batch normalization (BN), and rectified linear unit (ReLU) layers. At several convolutional layers, we visualize the image so that its features at the convolutional layer best match the features produced for an input image. The image produced from the features of the early layers tends to preserve the content of the input image, while it starts to show high similarity with the noiseprint of the input image as we go deeper in the network. Please note that the images are manipulated for the best visualization. The figure best viewed electronically with large zoom.

with the input image and start to show high similarity with the noiseprint of the input image. Thus, the features at the early and end layers can provide content and a noiseprint representations, respectively, of an image in the feature space.

These content and noiseprint representations are employed in both the optimization-based approach and the training of the noiseprint-injector. Specifically, we utilize loss functions that penalize both approaches' differences in content and noiseprint representations in the feature space. We choose the L2 loss because it is a convex and continuously-differentiable function. This allows the gradient-based optimization to find the global minimum. Please note that the original noiseprint extraction method (*Cozzolino & Verdoliva, 2020*) employed cross-entropy loss for the output of the binary classifier as shown in Fig. 2. This is because the goal is to encourage the Siamese architecture to generate a similar output for positive class patches and a different output for negative class patches. However, this is different from the goal of our loss which penalizes the differences in content and noiseprint deep representations. The following two subsections present the two proposed approaches in detail.

## Proposed optimization based neural noiseprint transfer approach

The proposed optimization-based approach is sketched in Fig. 5. The approach estimates the generated image $\mathbf{I}_g$ as

$$\mathbf{I}_g = \arg \min_{\mathbf{I}} \ell(\mathbf{I}_f, \mathbf{I}_a, \mathbf{I}), \tag{2}$$

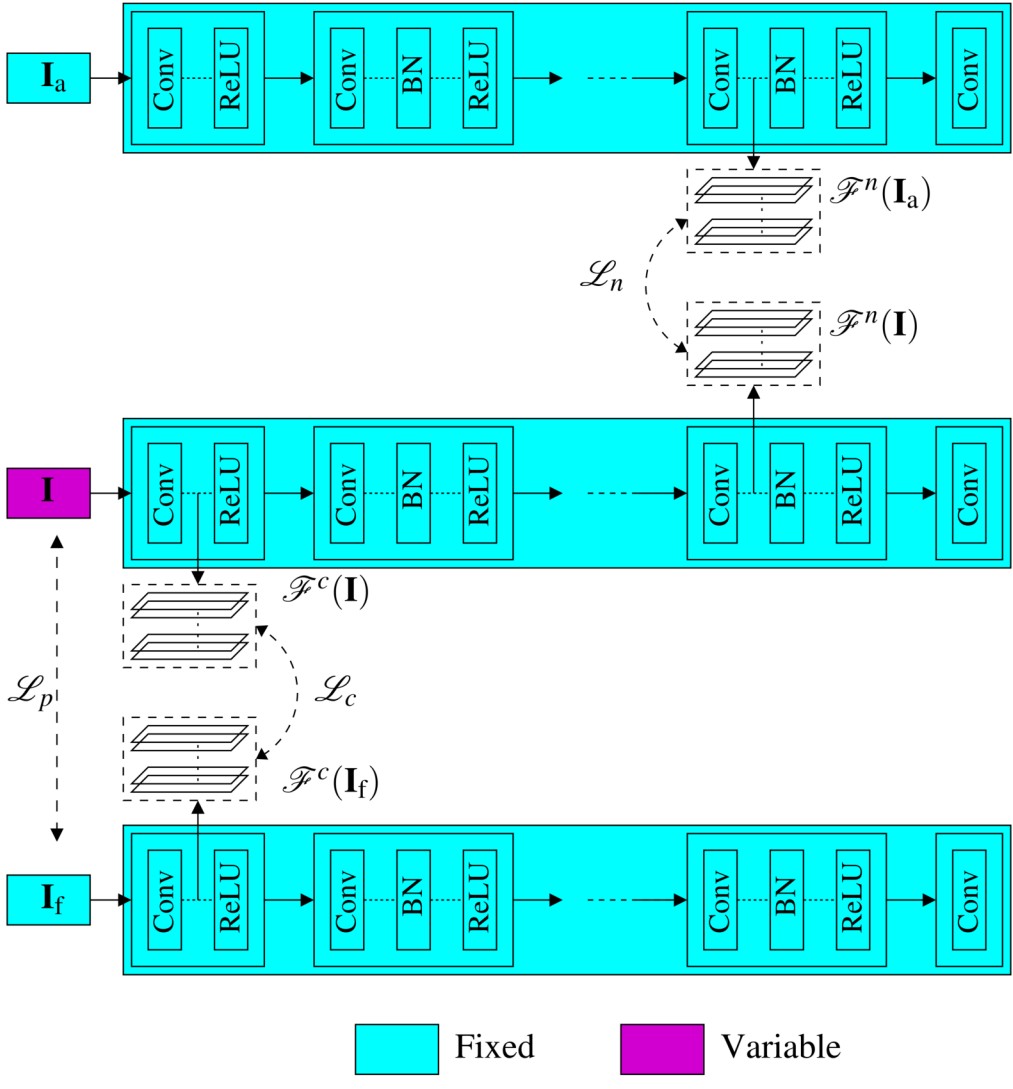

**Figure 5  Proposed optimization-based approach.** The approach finds an image $\mathbf{I}$ that minimizes the content representation loss $\mathcal{L}_c$, the noiseprint representation loss $\mathcal{L}_n$, and the pixel domain loss $\mathcal{L}_p$. Note that the weights of the original noiseprint extraction network are kept fixed, and the image $\mathbf{I}$ is the only learnable variable we need to optimize.

where

$$\ell(\mathbf{X}, \mathbf{Y}, \mathbf{Z}) = \alpha \mathcal{L}_c(\mathbf{X}, \mathbf{Z}) + \beta \mathcal{L}_n(\mathbf{Y}, \mathbf{Z}) + \gamma \mathcal{L}_p(\mathbf{X}, \mathbf{Z}) \tag{3}$$

is a loss function that comprises the following terms:

- The content loss $\mathcal{L}_c(\mathbf{X}, \mathbf{Z}) = \|\mathcal{F}^c(\mathbf{X}) - \mathcal{F}^c(\mathbf{Z})\|_2^2$ that penalizes the difference in content representations of $\mathbf{X}$ and $\mathbf{Z}$ at convolutional layer $c$.
- The noiseprint loss $\mathcal{L}_n(\mathbf{Y}, \mathbf{Z}) = \|\mathcal{F}^n(\mathbf{Y}) - \mathcal{F}^n(\mathbf{Z})\|_2^2$ that penalizes the difference in noiseprint representations of $\mathbf{Y}$ and $\mathbf{Z}$ at convolutional layer $n$.

- The pixel loss $\mathcal{L}_p(\mathbf{X}, \mathbf{Z}) = \|\mathbf{X} - \mathbf{Z}\|_2^2$ that penalizes the difference in pixel domain between $\mathbf{X}$ and $\mathbf{Z}$.

The factors $\alpha$, $\beta$, and $\gamma$ are weight-balancing factors between the different loss terms.

To solve Eq. (2), several numerical optimization techniques such as stochastic gradient descent or ADAM (*Kingma & Ba, 2015*) can be used, but our experiments show that the latter produces better results. In this case, we need to compute $\nabla \ell$, which is the gradient of the loss function $\ell$ w.r.t. $\mathbf{I}$. Note that the gradient of $\ell$ involves computing the differentiation of the deep feature representations obtained from different levels of the original noiseprint extraction network (*Cozzolino & Verdoliva, 2020*) as shown in Fig. 5. Luckily, these representations are typically composed of a chain of several linear and non-linear layers and are still differentiable. Additionally, since the optimization here is performed solely for the generated image (not the weights of the network), the network weights are kept fixed with the pre-trained weights from the original noiseprint extraction network. The detailed implementation of the proposed optimization-based approach is sketched in Algorithm 1.

Finally, as we solve Eq. (2) to estimate an image, the optimization is time-consuming as it involves a large number of computations, especially for large-size images. As we detail next, this problem is mitigated in our other proposed injection-based approach.

---

**Algorithm 1:** Proposed optimization-based approach.

**Data:** $\mathbf{I}_f$ and $\mathbf{I}_a$
**Result:** $\mathbf{I}_g$

1   $\mathbf{N}_a \leftarrow \mathcal{F}^n(\mathbf{I}_a)$ /* *get noiseprint representation of* $\mathbf{I}_a$*/ ;
2   $\mathbf{C}_f \leftarrow \mathcal{F}^c(\mathbf{I}_f)$ /* *get content representation of* $\mathbf{I}_f$*/ ;
3   $\mathbf{I} \leftarrow \mathbf{0}$;
4   Epoch $\leftarrow 0$;
5   **while** *Epoch is less than the total required number of epochs* **do**
6      $\mathbf{N}_i \leftarrow \mathcal{F}^n(\mathbf{I})$ /* *get noiseprint representation of* $\mathbf{I}$*/ ;
7      $\mathbf{C}_i \leftarrow \mathcal{F}^c(\mathbf{I})$ /* *get content representation of* $\mathbf{I}$*/ ;
8      $\mathcal{L}_c \leftarrow \alpha \|\mathbf{C}_f - \mathbf{C}_i\|_2^2$;
9      $\mathcal{L}_n \leftarrow \beta \|\mathbf{N}_a - \mathbf{N}_i\|_2^2$;
10      $\mathcal{L}_p \leftarrow \gamma \|\mathbf{I}_f - \mathbf{I}\|_2^2$;
11      $\mathcal{L} \leftarrow \mathcal{L}_c + \mathcal{L}_n + \mathcal{L}_p$;
12      Perform one update step for $\mathbf{I}$ using the ADAM optimizer by minimizing $\mathcal{L}$ w.r.t. $\mathbf{I}$;
13      epoch $\leftarrow$ epoch + 1;
14   $\mathbf{I}_g \leftarrow \mathbf{I}$;

---

## Proposed injection based neural noiseprint transfer approach

Instead of optimizing for the generated image, which is time-consuming for large images, as we presented in the previous subsection, here we propose another much faster approach. Specifically, we propose a novel noiseprint-injector neural network that learns how to transfer the noiseprint of an image into another one. The training of the network is

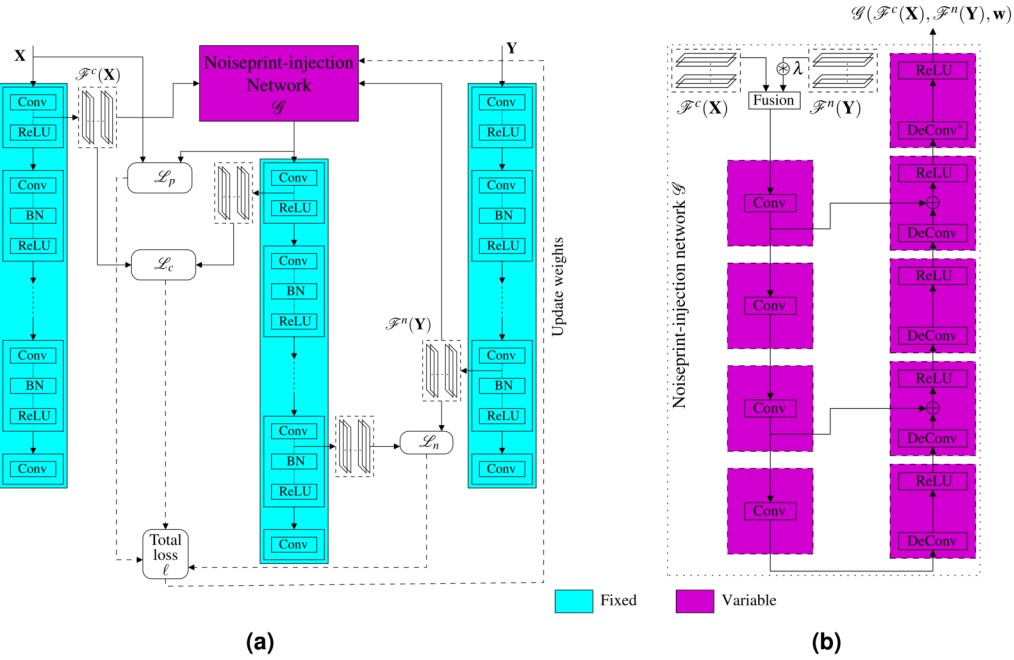

**Figure 6  Proposed injection-based neural noiseprint transfer approach.** As shown in (A), the noiseprint-injector is trained to take the content, and noiseprint representations of two different images $\mathbf{X}$ and $\mathbf{Y}$, and produce another image $\mathcal{G}(\mathcal{F}^c(\mathbf{X}), \mathcal{F}^n(\mathbf{Y}), \mathbf{w})$ that has similar content representations and pixel values with $\mathbf{X}$, while has similar noiseprint representations with $\mathbf{Y}$. In (B), we present the proposed noiseprint-injector network architecture. Note that the weights of the original noiseprint extraction network are kept fixed, and the weights of the proposed noiseprint-injector network are the learnable variables we need to optimize.

performed offline and completely in an unsupervised fashion. Let the noiseprint-injector network be denoted as $\mathcal{G}$ and parameterized by the parameter vector $\mathbf{w}$. As shown in Fig. 6A, the noiseprint-injector is trained to take the content and noiseprint representations of two different images $\mathbf{X}$ and $\mathbf{Y}$, and produce another image that has:

- Similar content representations in the feature space with the first image $\mathbf{X}$.
- Similar values in pixel domain with $\mathbf{X}$.
- Similar noiseprint representations in the feature space of the second image $\mathbf{Y}$.

In other words, the goal of training the noiseprint-injector network is to make the network learns how to produce an output image that bears the noiseprint of the image $\mathbf{Y}$ while looking visually similar to $\mathbf{X}$. This goal can be achieved by estimating the best parameter vector $\mathbf{w}^*$ of the noiseprint-injector network that minimizes the loss $\ell$ defined in Eq. (3). Mathematically,

$$\mathbf{w}^* = \arg \min_{\mathbf{w}} \sum_{\forall \mathbf{X} \neq \mathbf{Y}} \ell(\mathbf{X}, \mathbf{Y}, \mathcal{G}(\mathcal{F}^c(\mathbf{X}), \mathcal{F}^n(\mathbf{Y}), \mathbf{w})). \tag{4}$$

The training procedure of the proposed noiseprint-injector network (shown in Fig. 6A) comprises the following steps.[1] At each learning epoch, we first get the content

[1]Keep in mind that the actual implementation is performed on patches, not single images.

representation of $\mathbf{X}$ and the noiseprint representation of $\mathbf{Y}$. Then we perform a forward pass of the noiseprint-injector network to get the output image. Then, we obtain both the content and the noiseprint representations of the output image and form the total loss in Eq. (3) with these representations in addition to the representations extracted from $\mathbf{X}$ and $\mathbf{Y}$. Finally, we perform a backward optimization step for the noiseprint-injector network using the ADAM optimizer to update the parameters $\mathbf{w}$. The detailed implementation of these steps is presented in Algorithm 2. After estimating $\mathbf{w}^*$, the noiseprint-injector can produce the generated image from the authentic and forged images as

$$\mathbf{I}_g = \mathcal{G}(\mathcal{F}^c(\mathbf{I}_f), \mathcal{F}^n(\mathbf{I}_a), \mathbf{w}^*). \tag{5}$$

Specifically, we get the content representation of $\mathbf{I}_f$ and the noiseprint representation of $\mathbf{I}_a$ and feed them to the noiseprint-injector network (with its weights fixed to $\mathbf{w}^*$) to produce $\mathbf{I}_g$.

The architecture of the proposed noiseprint-injector network is shown in Fig. 6B. The details of the input–output sizes, kernel, and stride lengths of each layer are shown in Table 1. First, we combine the input content and noise representations in the feature space by stacking them.[2] Because the content and noise representations differ in the dynamic range, we multiply the noise representations by a factor $\lambda$ before combing them. Then, the stacked features are presented to a convolution deconvolution auto-encoder network with symmetric skip connections. A ReLU activation layer follows each deconvolution layer. Our noiseprint-injector network is similar to the network proposed in *Mao, Shen & Yang (2016)* for image restoration. However, our network here is much shallower, and the inputs to the network are feature space representations, not an image as in *Mao, Shen & Yang (2016)*.

We emphasize that the goal of training our noiseprint-injector neural network is to learn how to transfer the noiseprint of an image into another one. There is no **restriction** about the image-pair used for the training. The only requirement of the image pair is to contain images with different noiseprints so the network can learn how to transfer the noiseprint from one image into another while generating a high-fidelity image. This requirement of different noiseprints can be easily fulfilled by just using images obtained from different cameras or from the same camera but cropped at different spatial locations. This again enforces our claim that the proposed noiseprint-injection approach is generic and does not require any sophisticated datasets (that are not easy to obtain) in training. After training, the generated image is estimated in one forward pass of the network without any optimization. Therefore, the proposed noiseprint injection-based approach is much faster than the proposed optimization-based approach discussed in the previous subsection. Also, as the proposed architecture does not include any size-changing layer (such as pooling), the architecture works for any input size ($a \times b$). In other words, there is no need to split the input image into non-overlapping windows with specific sizes to make it suitable for the network.

[2] Note that the number of channels of the content or noise representations is 64. This is in accordance with the output of the convolutional layers in *Zhang et al. (2017)*.

**Table 1  Noiseprint-injector network architecture.** K stands for kernel size, and S for stride. The architecture is suitable for any input image size $(a \times b)$.

| Layer | InSize | K | S | OutSize |
|---|---|---|---|---|
| Conv | $a \times b \times 128$ | 3 | 1 | $a \times b \times 128$ |
| DeConv | $a \times b \times 128$ | 3 | 1 | $a \times b \times 128$ |
| DeConv * | $a \times b \times 128$ | 3 | 1 | $a \times b \times 1$ |

---

**Algorithm 2:** Illustration of the proposed noiseprint-injector network training procedure.

---

**Data:** Dataset that contains images with different noiseprints

**Result:** Best parameter vector $\mathbf{w}^*$

1 Epoch $\leftarrow 0$;

2 Initialize the parameter vector $\mathbf{w}$;

3 **while** *Epoch is less than the total required number of epochs* **do**

4     Get two different images $\mathbf{X}$ and $\mathbf{Y}$ from the dataset;

5     $\mathbf{C}_x \leftarrow \mathcal{F}^c(\mathbf{X})$ /* get content representation of $\mathbf{X}$*/ ;

6     $\mathbf{N}_y \leftarrow \mathcal{F}^n(\mathbf{Y})$ /* get noiseprint representation of $\mathbf{Y}$*/ ;

7     $\mathbf{I} \leftarrow \mathcal{G}(\mathbf{C}_x, \mathbf{N}_y, \mathbf{w})$ /* forward pass to generate an output from the proposed noiseprint-injector network */ ;

8     $\mathbf{C}_i \leftarrow \mathcal{F}^c(\mathbf{I})$ /* get content representation of $\mathbf{I}$*/ ;

9     $\mathbf{N}_i \leftarrow \mathcal{F}^n(\mathbf{I})$ /* get noiseprint representation of $\mathbf{I}$*/ ;

10     $\mathcal{L}_c \leftarrow \alpha \|\mathbf{C}_x - \mathbf{C}_i\|_2^2$;

11     $\mathcal{L}_n \leftarrow \beta \|\mathbf{N}_y - \mathbf{N}_i\|_2^2$;

12     $\mathcal{L}_p \leftarrow \gamma \|\mathbf{X} - \mathbf{I}\|_2^2$;

13     $\mathcal{L} \leftarrow \mathcal{L}_c + \mathcal{L}_n + \mathcal{L}_p$;

14     Perform one update step for $\mathbf{w}$ using the ADAM optimizer by minimizing $\mathcal{L}$ w.r.t. $\mathbf{w}$ (backward pass);

15     epoch $\leftarrow$ epoch + 1;

16 $\mathbf{w}^* \leftarrow \mathbf{w}$;

---

# EXPERIMENTAL ANALYSIS

The effectiveness of the proposed counter-forensic approaches is evaluated through extensive experimental analysis on attacking two noiseprint-based forensics methods. The first is the original noiseprint method (*Cozzolino & Verdoliva, 2020*) which denoted as *OrgNoiseprint*, and the other is the method proposed in *Mareen et al. (2022)* which denoted as *Comprint*. The OrgNoiseprint method utilizes the noiseprint only in the forensic tasks, while the Comprint method blends the noiseprint with the compression fingerprints to boost the performance of the forensic tasks. Additionally, since there are no known counter-forensic methods for the noiseprint specifically, the proposed counter-forensic approaches are compared with the generic counter-forensic median filter-based denoising

approach. In our evaluation, we focus on the attacks for two popular forensic applications: the forgery localization and the camera model identification applications.

In the rest of this section, we first present the training procedure of the proposed noiseprint-injector network and the specific values of the parameters used to report our experimental results for both proposed approaches. Then, we present the two forensic attacks in detail, showing the used datasets and the evaluation metrics for each attack. Finally, we discuss our final thoughts at the end of this section.

## Training and parameters setting
### Optimization based approach

As shown in Fig. 5, the optimization-based approach tries to answer this simple question: what is the image that has similar content and pixel representations with a forged image and simultaneously has a similar noiseprint representation with an authentic image? Thus, the optimization is performed solely for the image, not the network parameters, which means it works out of the box without training. Specifically, the approach estimates the generated image by minimizing the loss in Eq. (2) with respect to the generated image. Since the content and noiseprint deep representations are obtained from different levels of the original noiseprint extraction network (Cozzolino & Verdoliva, 2020) as indicated in 'Deep feature representations', the functions of these layers are involved in the optimization. However, the network weights are kept fixed with the pre-trained weights from the original noiseprint extraction method (Cozzolino & Verdoliva, 2020) because the optimization is performed solely for the generated image (not the weights).

### Training of noiseprint-injector network

As indicated in 'Proposed injection based neural noiseprint transfer approach', the only restriction about image pairs used for the training is that the image pair must contain images with different noiseprints. This can be easily fulfilled by just using images obtained from different cameras or obtained from the same camera but cropped at different spatial locations. The training dataset comprises several image patches cropped at random locations from different image pairs obtained from the Dresden dataset (Gloe & Böhme, 2010). Specifically, we used a total number of 10,000 patches. The size of each patch is $256 \times 256$. The patches are obtained by cropping the images of the Dresden dataset at random locations. Then, the 10,000 patches are shuffled and then introduced to the noiseprint-injector neural network for training.

### Training settings

The training was performed on an NVIDIA® Tesla® V100 GPU installed in a Ubuntu machine with 128 GB RAM. The parameters are set as $c = 1$ and $n = 59$ for both proposed approaches in all experiments. The number of epochs is set to 300. We set the hyper-parameters for the noiseprint injection-based approach as $\alpha = 1$, $\beta = 1$, $\gamma = 10000$, $\lambda = 0.01$, and the learning rate $= 10^{-5}$. The hyper-parameters for the optimization-based approach are set to be $\alpha = 10$, $\beta = 1$, $\gamma = 10$, and the learning rate $= 0.006$. We used a smaller sample from our training/test dataset with the grid search approach to select the best-performing hyper-parameters. The best-performing parameters here mean the

parameters that give the best performance on the test set from the point of view of attacking the noiseprint-based forensics based on several forensic accuracy scores that are discussed in the next subsection. Note that only the luminance channel is provided for the proposed approaches for input color images. The output colored image is obtained by combining the output luminance channel with the CbCr channels of the input image.

## Forgery localization attack

We evaluate the effectiveness of the proposed counter-forensic approaches in attacking the OrgNoiseprint and the Comprint forgery localization methods.[3] Additionally, the performance of the proposed approaches is compared with the median filtering-based approach. We used both $3 \times 3$ and $5 \times 5$ kernel sizes for the median based-approaches. All counter-forensic approaches under comparison (the proposed and the median filter approaches) are applied to forged images to produce newly generated images. Then, the forgery is localized in the generated images by searching for inconsistencies in their noiseprints using the Splicebuster blind localization algorithm (*Cozzolino, Poggi & Verdoliva, 2015*). The output of the localization algorithm is a real-valued heat map that indicates, for each pixel, the likelihood that it has been forged.

To provide a quantitative evaluation of the counter-forensic approaches under comparison, we evaluate them in two aspects. First, we need to measure the degree of visual similarity between the forged images and the generated ones. The PSNR and the SSIM (*Wang et al., 2004*) measures are used for this purpose. Second, we treat forgery localization as a binary classification problem in which pixels of the generated image can either belong to forged or authentic classes. Then, we employ several measures to evaluate the effect of the approaches under comparison on the performance of this binary classification results. All employed measures are built on four fundamental quantities, which are: (a) TP (true positive): # forged pixels declared forged, TN (true negative): # authentic pixels declared authentic, (c) FP (false positive): # authentic pixels declared forged, and (d) FN (false negative): # forged pixels declared authentic. From these four quantities, we measure the F1 score, Matthews Correlation Coefficient (MCC), and average precision (AP) as in *Cozzolino & Verdoliva (2020)*. Note that, as in *Cozzolino & Verdoliva (2020)*, the heat map generated by the localization algorithm is thresholded. Then, the F1 and MCC scores are reported as the maximum over all possible thresholds. Also, the AP is computed as the area under the precision–recall curve. Three datasets are used in our evaluation of the counter forgery localization, which are the DSO-1 dataset (*de Carvalho et al., 2013*), Decision-Fusion dataset (*Fontani et al., 2013*), and the Korus dataset (*Korus & Huang, 2017*). Each dataset has different characteristics. The DSO-1 dataset is characterized by large splicings and uncompressed images. The Korus dataset contains raw images (not JPEG compressed) with several forgeries, such as copy-move and object removals. The Decision-Fusion dataset mimics a realistic scenario of professional forgery and saving the forged images in JPEG compressed format. Note that the input to the proposed approaches is a forged image from a dataset with a corresponding same-size authentic image selected randomly. In contrast, the median filtering approach's input is only a forged image.

[3]The source code of both methods is publicly available online.

Tables 2 and 3 show the quantitative evaluation of the counter-forensic approaches under comparison in attacking the OrgNoiseprint and the Comprint forgery localization methods, respectively. Specifically, for each dataset, the average F1 score, MCC score, and AP values are reported for the proposed optimization-based, the proposed noiseprint injection-based, and the median filtering-based approaches. We also report the average PSNR and SSIM between the generated images and the corresponding forged ones. To verify the consistency of the performance of the proposed approaches w.r.t. the random selection of the authentic images, the experiment is repeated ten times for each forged image. As shown in the tables, the proposed approaches significantly reduce the values of the used metrics (F1, MCC, and AP) compared with the original values associated with the OrgNoiseprint and the Comprint forgery localization methods. Also, the proposed approaches produce better metrics values compared with the median filtering-based approach. However, the median based-approach with $5 \times 5$ kernel produces better results than the one with $3 \times 3$ kernel but at the expense of more degradation to the visual fidelity. At the same time, the generated images by the proposed approaches show high visual fidelity to the forged ones, as reported by the PSNR and SSIM average values. We can see that the proposed approaches reduce the forensic accuracy scores for the DSO-1 dataset by an average of 75% compared with the OrgNoiseprint method while at the same time keeping an average PSNR of 31.5 dB and SSIM of 0.9.

The numerical results shown in the tables are reinforced by presenting some visual results of the OrgNoiseprint approach, the proposed optimization-based approach, and the proposed injection-based approach in Figs. 7, 8, and 9, respectively. In each row in Fig. 7, the columns from left to the right show the forged image, its extracted noiseprint, and its heat map. Similarly, in each row in Figs. 8 and 9, the columns from left to the right show the generated image, the noiseprint of the generated image, and its heat map. Below each row, we report the values of the F1 score, MCC score, and AP value. Again, the reported values show the effectiveness of the proposed approaches, which significantly reduces the values of the used metrics (F1, MCC, and AP) compared with the original values associated with the OrgNoiseprint method.

Please note that despite the median filtering based-approach producing fair results, the approach is vulnerable to easy detection (*Kirchner & Fridrich, 2010*; *Cao et al., 2010*; *Zhu, Gu & Chen, 2022*). Based on the median filtering characteristics, several hand-crafted features are designed and used along with machine learning algorithms for the detection of the application of the median filter on a digital image. Examples of the employed features from the filtered image include the gradient between neighboring pixels, the Local Binary Patterns (*Zhu, Gu & Chen, 2022*), the Fourier Transform coefficients (*Rhee, 2015*), and the singular values (*Amanipour & Ghaemmaghami, 2019*). In contrast, the proposed approaches synthesize a newly generated image that is visually imperceptible to a forged image, but its noiseprint is very close to the noiseprint of an authentic image. Thus, the proposed approaches do not leave statistical traces that can be detected, as in the case of the median filtering-based approach.

A final note we want to highlight is that the proposed approaches are designed to attack the forensics approaches that are built on the utilization of the noiseprint. So, the

**Table 2 Quantitative evaluation of the proposed approaches and the median filtering based approach in attacking the OrgNoiseprint method (*Cozzolino & Verdoliva, 2020*).** Note that we use the notation $x(y)$ to report the values in the table, where $x$ represents the value of a metric and $y$ represents the reduction percentage compared with its original value (before applying the counter-forensic approach). Bold values represent the highest reduction percentage in each dataset and metric.

| | Metrics | Datasets | | |
|---|---|---|---|---|
| | | DSO-1 | Decision-Fusion | Korus |
| OrgNoiseprint | F1 | 0.802 | 0.520 | 0.345 |
| | MCC | 0.796 | 0.527 | 0.347 |
| (*Cozzolino & Verdoliva, 2020*) | AP | 0.785 | 0.445 | 0.298 |
| | F1 | 0.279 (**65%**) | 0.189 (**64%**) | 0.146 (**57%**) |
| Proposed | MCC | 0.181 (77%) | 0.185 (65%) | 0.136 (**60%**) |
| optimization | AP | 0.218 (**72%**) | 0.149 (**67%**) | 0.097 (**67%**) |
| approach | PSNR | 31.39 | 31.47 | 32.53 |
| | SSIM | 0.896 | 0.899 | 0.906 |
| | F1 | 0.288 (64%) | 0.196 (62%) | 0.151 (56%) |
| Proposed | MCC | 0.176 (**77%**) | 0.180 (**66%**) | 0.140 (59%) |
| noiseprint-injection | AP | 0.222 (71%) | 0.155 (65%) | 0.104 (65%) |
| approach | PSNR | 31.52 | 31.40 | 31.45 |
| | SSIM | 0.901 | 0.891 | 0.881 |
| | F1 | 0.368 (54%) | 0.224 (56%) | 0.191 (44%) |
| Median (3 × 3) | MCC | 0.279 (65%) | 0.251 (52%) | 0.186 (46%) |
| filtering-based | AP | 0.304 (61%) | 0.168 (62%) | 0.145 (51%) |
| approach | PSNR | 32.51 | 32.92 | 34.06 |
| | SSIM | 0.942 | 0.932 | 0.944 |
| | F1 | 0.315 (60%) | 0.217 (58%) | 0.178 (48%) |
| Median (5 × 5) | MCC | 0.230 (71%) | 0.242 (54%) | 0.179 (48%) |
| filtering-based | AP | 0.255 (67%) | 0.152 (65%) | 0.131 (55%) |
| approach | PSNR | 30.88 | 30.63 | 30.80 |
| | SSIM | 0.899 | 0.869 | 0.866 |

performance of the proposed approaches is closely tied to the success of the noiseprint-based forensics approaches. If the original noiseprint-based forensics approaches do not detect a specific type of forgery, the proposed approaches will not succeed accordingly. This is clear from the reported results in Tables 2 and 3 for the Korus dataset. The original noiseprint-based forensics approaches do not behave well for this dataset because the dataset constitutes raw images (not JPEG compressed) while noiseprintis trained on JPEG-compressed images and also because the dataset contains several contemporary forgeries.

## Camera model identification attack

Here, we evaluate the effect of the proposed approaches and the median filter approach on the camera model identification application. Specifically, the IEEE Forensic Camera Model Identification Challenge hosted on the Kaggle platform (https://www.kaggle.com/competitions/sp-society-camera-model-identification) is used for this purpose. Five camera models are used from the dataset: Sony-NEX-7, HTC-1-M7,

**Table 3 Quantitative evaluation of the proposed approaches and the median filtering based approach in attacking the Comprint method (*Mareen et al., 2022*).** Bold values represent the highest reduction percentage in each dataset and metric.

| | Metrics | Datasets | | |
|---|---|---|---|---|
| | | DSO-1 | Decision-Fusion | Korus |
| Comprint | F1 | 0.804 | 0.551 | 0.287 |
| | MCC | 0.808 | 0.549 | 0.301 |
| (*Mareen et al., 2022*) | AP | 0.791 | 0.456 | 0.228 |
| Proposed optimization approach | F1 | 0.295 (**63%**) | 0.211 (**62%**) | 0.169 (41%) |
| | MCC | 0.211 (**74%**) | 0.208 (62%) | 0.171 (43%) |
| | AP | 0.221 (**72%**) | 0.149 (**67%**) | 0.119 (48%) |
| | PSNR | 32.12 | 31.53 | 31.85 |
| | SSIM | 0.908 | 0.867 | 0.893 |
| Proposed noiseprint-injection approach | F1 | 0.307 (61%) | 0.213 (61%) | 0.168 (**41%**) |
| | MCC | 0.215 (73%) | 0.206 (**62%**) | 0.167 (**45%**) |
| | AP | 0.234 (70%) | 0.151 (66%) | 0.115 (**50%**) |
| | PSNR | 31.42 | 31.43 | 31.79 |
| | SSIM | 0.897 | 0.861 | 0.890 |
| Median (3 × 3) filtering-based approach | F1 | 0.343 (57%) | 0.218 (60%) | 0.190 (33%) |
| | MCC | 0.299 (63%) | 0.247 (55%) | 0.205 (31%) |
| | AP | 0.291 (63%) | 0.157 (65%) | 0.144 (36%) |
| | PSNR | 32.51 | 32.92 | 34.06 |
| | SSIM | 0.942 | 0.932 | 0.944 |
| Median (5 × 5) filtering-based approach | F1 | 0.306 (61%) | 0.215 (61%) | 0.178 (37%) |
| | MCC | 0.255 (68%) | 0.238 (56%) | 0.198 (34%) |
| | AP | 0.251 (68%) | 0.151 (67%) | 0.131 (42%) |
| | PSNR | 30.88 | 30.63 | 30.83 |
| | SSIM | 0.899 | 0.869 | 0.866 |

iPhone-6, LG-Nexus-5x, and Samsung-Galaxy-Note 3. As in *Cozzolino & Verdoliva (2020)*, 100 training images are used from each camera to estimate the noiseprint of the camera by averaging the noiseprint extracted from each training image. Then, we crop the central $500 \times 500$ of 100 different test images from each camera to be used in the identification. A test image is related to a camera if the Euclidean distance from its noiseprint to the average noiseprint of the camera is minimum.

As we discussed, the input to our proposed approaches is an image pair. The approaches produce a generated image visually similar to the first input image while having similar noiseprint with the second one. Thus, to apply the proposed approaches in this attack, we form several input image pairs. In each pair, the first image is a test image obtained from a specific camera in the dataset, while the second is taken at random from the test images of a different camera.

Figure 10 shows the confusion matrices obtained for the camera model identification using: (a) the original noiseprint forensics method (*Cozzolino & Verdoliva, 2020*), (b) after applying the proposed optimization-based approach, (c) after applying the proposed noiseprint-injection based approach, and (d) after applying the median-filter based

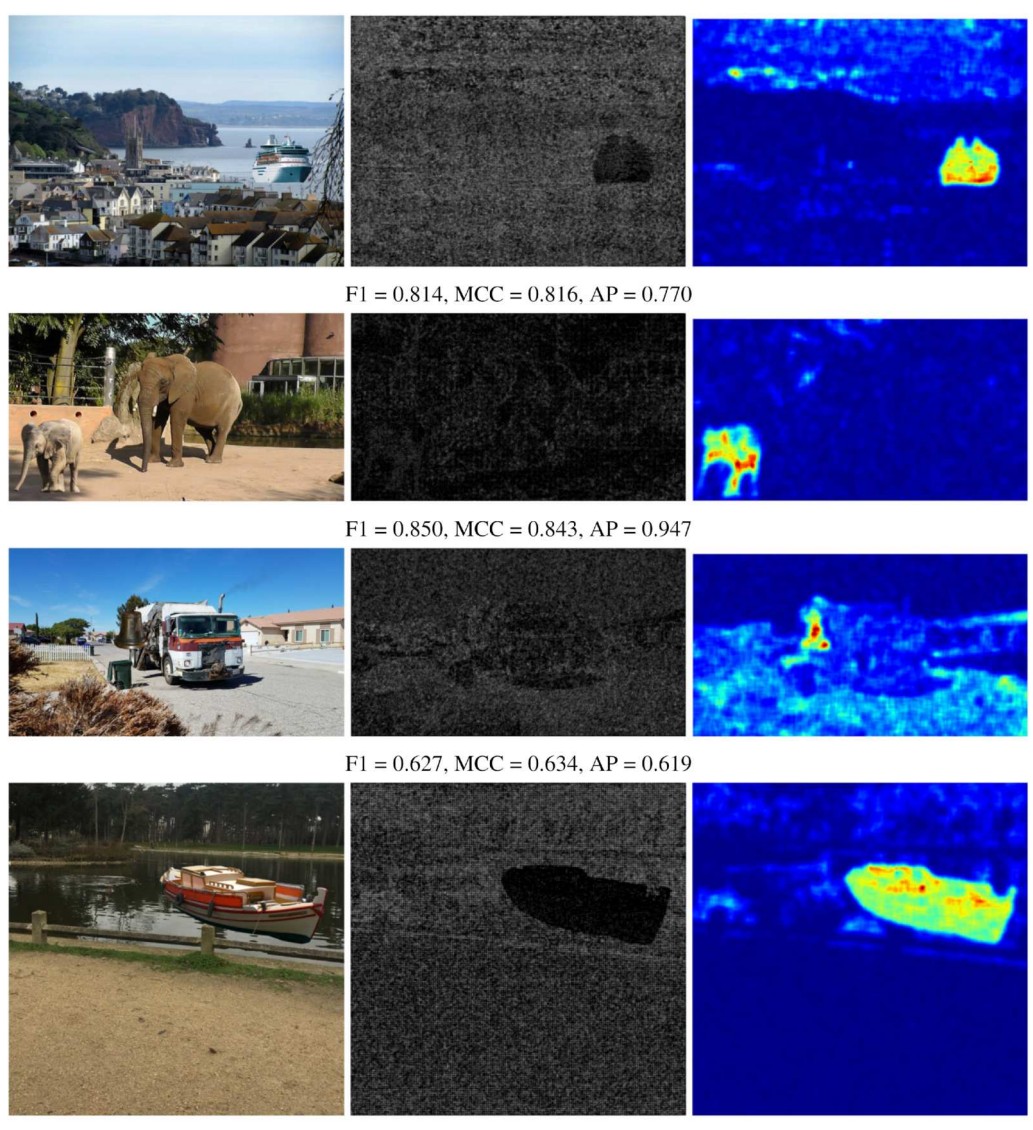

F1 = 0.814, MCC = 0.816, AP = 0.770

F1 = 0.850, MCC = 0.843, AP = 0.947

F1 = 0.627, MCC = 0.634, AP = 0.619

F1 = 0.941, MCC = 0.938, AP = 0.933

**Figure 7** **Visual results of the OrgNoiseprint approach (***Cozzolino & Verdoliva, 2020***) for example images.** The columns from left to right show: the forged image, its noiseprint, and its heat-map. The values of the F1 score, MCC score, and AP value for each image are reported below each row. The figure best viewed electronically with large zoom.

approach. As shown in the figure, the proposed approaches can significantly reduce the accuracy of the confusion matrix, which means that the proposed approaches are very effective in camera model identification attacks.

## Discussion

In this section, we present several discussion points about the proposed approaches. First, the execution time of the proposed approaches is discussed. Then, the success of the

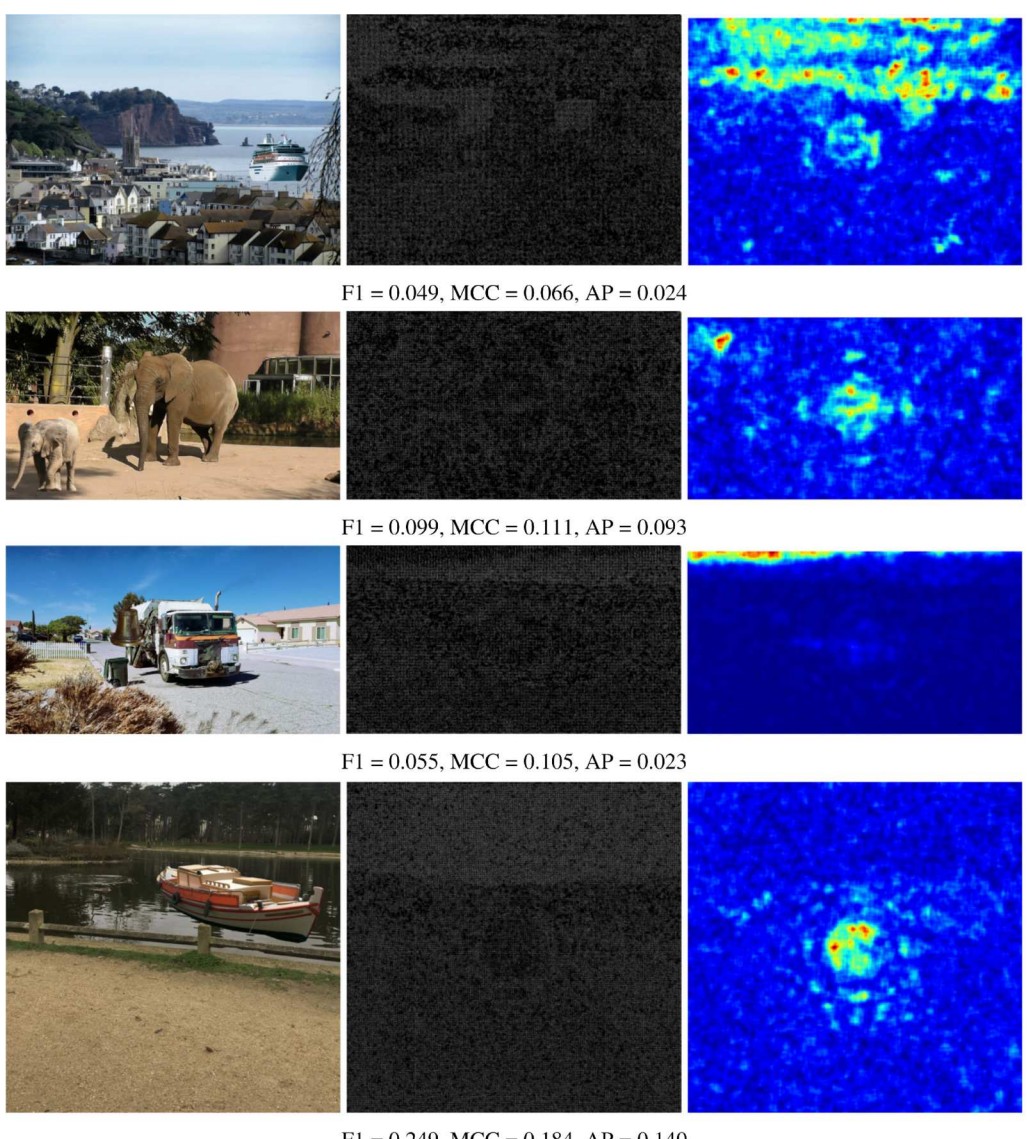

F1 = 0.049, MCC = 0.066, AP = 0.024

F1 = 0.099, MCC = 0.111, AP = 0.093

F1 = 0.055, MCC = 0.105, AP = 0.023

F1 = 0.249, MCC = 0.184, AP = 0.140

**Figure 8** **Visual results of the proposed optimization-based approach for example images.** The columns from left to right show: the generated image by the proposed optimization-based approach, the noiseprint of the generated image, and its heat-map. The figure best viewed electronically with large zoom.

proposed approaches in transferring the noiseprint of the authentic images to the generated ones is highlighted. Then, the consistency of the performance of the proposed approaches is verified against the different selections of authentic input images. Then, we highlight the similarity between the proposed and other existing approaches dedicated to artistic style transfer. Finally, some future research directions are presented.

### Execution time

As presented in the previous two subsections, the proposed approaches are already evaluated on different datasets. Each dataset contains images with different resolutions,

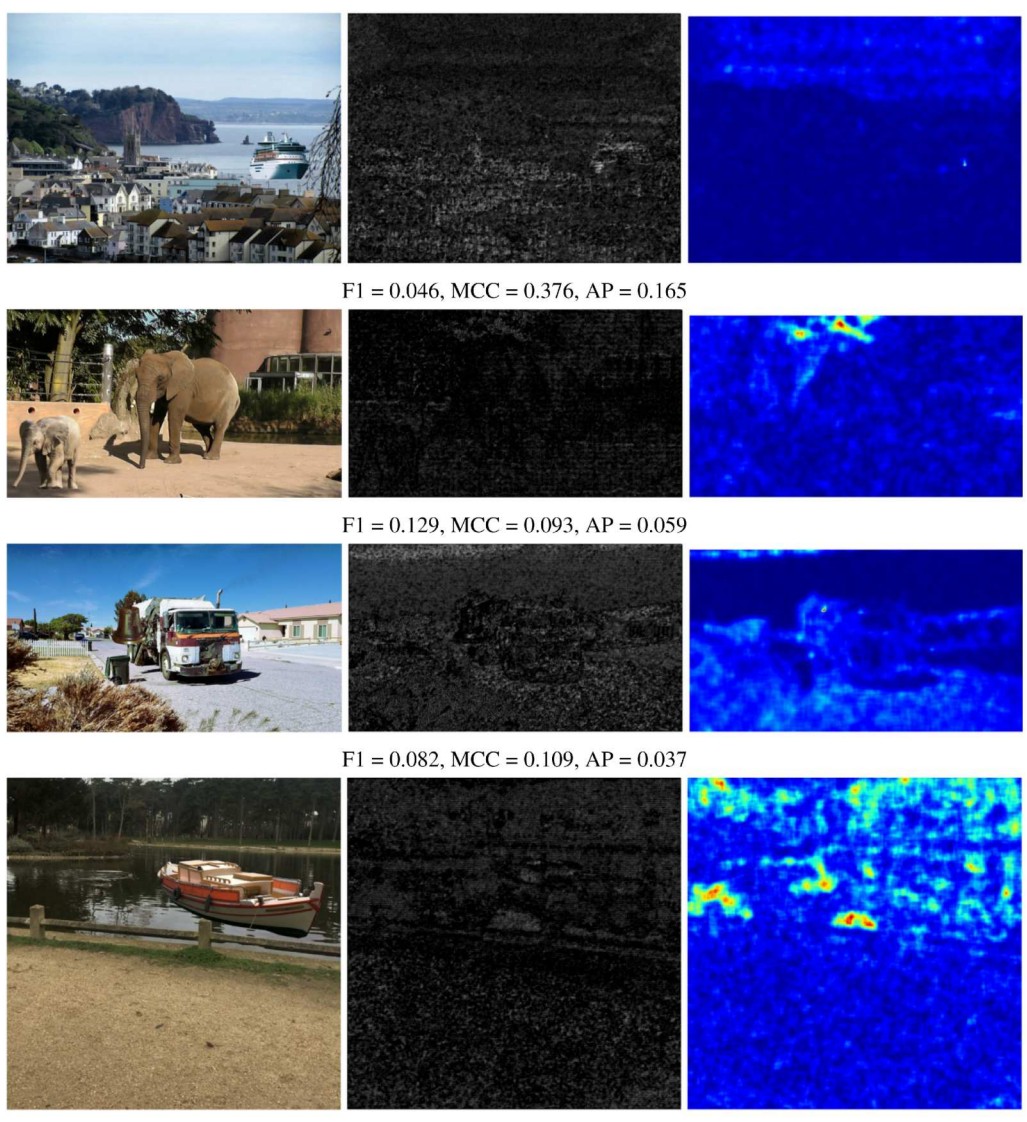

F1 = 0.046, MCC = 0.376, AP = 0.165

F1 = 0.129, MCC = 0.093, AP = 0.059

F1 = 0.082, MCC = 0.109, AP = 0.037

F1 = 0.254, MCC = 0.231, AP = 0.131

**Figure 9** **Visual results of the proposed noiseprint-injection based approach for example images.** The columns from left to right show: the generated image by the proposed noiseprint-injection based approach, the noiseprint of the generated image, and its heat-map. The figure best viewed electronically with large zoom.

types, and sizes. This reinforces our claim that the proposed approaches are generic and do not tie to specific images or camera models. Additionally, there is no or little difference in performance between the proposed optimization-based approach and noiseprint-injection-based approach. However, the latter is much faster compared to the former. Specifically, for the exact resolution of newly generated images, the proposed noiseprint-injection-based approach is about 100× faster than the optimization-based approach in generating the image. For example, for images with HD (1280 × 720) resolution, the optimization-based

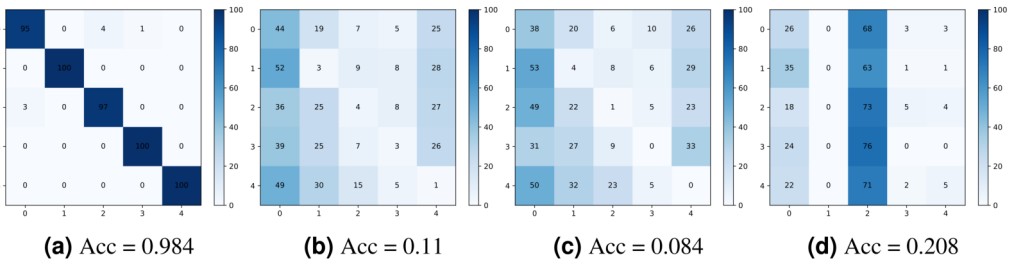

**(a)** Acc = 0.984     **(b)** Acc = 0.11     **(c)** Acc = 0.084     **(d)** Acc = 0.208

**Figure 10** Confusion matrices were obtained for the camera model identification using: (A) the original noiseprint forensic method (*Cozzolino & Verdoliva, 2020*), (B) after applying the proposed optimization-based approach, (C) after applying the proposed noiseprint -injection based approach, and (D) after applying the median-filter based approach.

approach produces the generated image in about 10 min on the hardware specified in the experimental analysis section. This time can be reduced if the machine is equipped with more than one GPU. On the other hand, the noiseprint-injection approach produces the generated image for the same resolution on the same hardware configuration in about 5 seconds. This is because the noiseprint-injection-based approach performs only a simple forward pass when generating the new image. However, the noiseprint-injection-based approach requires training the noiseprint-injector network, while the optimization-based approach works out of the box without any training. In other words, the noiseprint-injection-based approach shifts the image generation time from the generation phase to the training phase, saving significant time in generating the new image. Nevertheless, the execution time of the noiseprint-injection-based approach is not that large compared with the original noiseprint extraction method, which produces the noiseprint for an image with HD resolution in about 3 seconds.

### Noiseprint transfer

To show the success of the proposed approaches in transferring the noiseprint from the authentic image to the generated one, we performed the same experiment in Table 2 and recorded the mean squared error of the noiseprint between the generated image and (a) the forged image (Generated *vs.* Forged) and (b) the authentic image (Generated *vs.* Authentic). Since we repeated the experiments in Table 2 ten times for each forged image, we used the box plot to represent the obtained (Generated *vs.* Forged) and (Generated *vs.* Authentic) values. These plots are shown for the first ten images[4] from the DSO-1 and Korus datasets in Figs. 11A and 11B, respectively. Also, a solid line is drawn in each figure to connect the average value of each box for better visualization. The upper and lower rows of the figures display the results for the optimization and the injection-based approaches, respectively.

As shown in the figure, the reported (Generated *vs.* Authentic) values are significantly lower than the (Generated *vs.* Forged) values. In other words, the noiseprint similarity between the generated and authentic images is significant compared with the noiseprint similarity between the generated images and the original forged ones. This means that the

[4]We obtained similar results for all other images but display the plots for ten images only for better visualization of the plots.

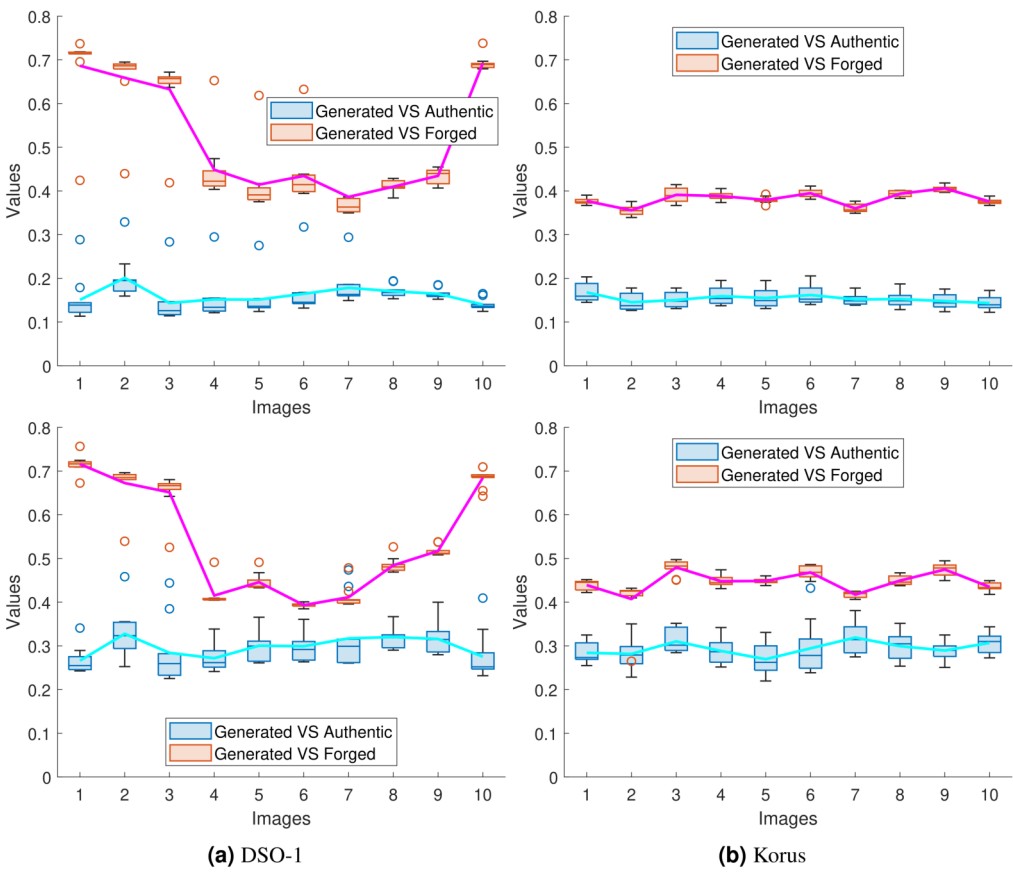

**(a)** DSO-1  **(b)** Korus

**Figure 11** **Box plots of the mean squared error of the noiseprint between the generated and the forged images (Generated *vs.* Forged); and between the generated and the authentic images (Generated *vs.* Authentic).** The plots shown in (A) and (B) are for the first 10 images in the DSO-1 and Korus datasets, respectively. The upper and lower rows display the results for the optimization and the injection based approaches, respectively.

proposed approaches successfully transfer the noiseprint from the authentic image to the forged one.

### Performance consistency

Since the input to the proposed approaches is a forged image from a dataset with a corresponding same-size authentic image selected randomly, we need to verify the consistency of the performance with the different selections of authentic images. In Tables 2 and 3, we repeated the experiments ten times for each forged image and reported the average values. However, the average value does not indicate the spread of the reported numbers. Here, instead of reporting the average values only, we used the box plot to represent the obtained F1, MCC, and AP values for the proposed approaches for the first 10 images from the DSO-1 dataset. We show these box plots in Fig. 12.

From the box plots, we can see that the proposed approaches produce consistent (F1, MCC, AP) values regardless of the used authentic images, as shown from the spread of each

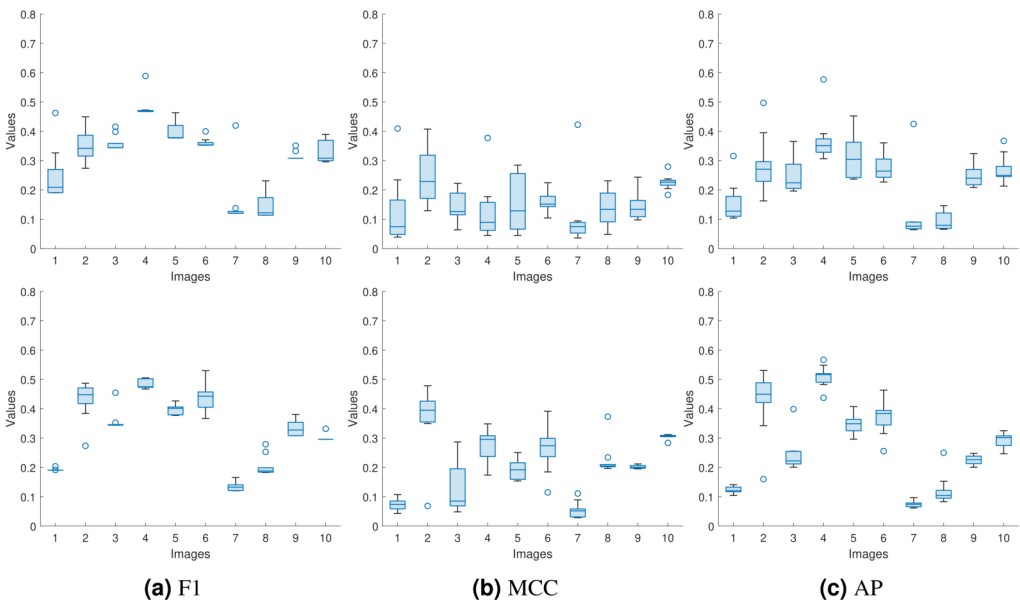

**(a)** F1        **(b)** MCC        **(c)** AP

**Figure 12** **Box plots for the (A) F1, (B) MCC, and (C) AP values for the first ten images in the DSO-1 dataset.** The upper and lower rows display the results for the optimization and the injection based approaches, respectively.

box of the ten images. This is because the approaches successfully transfer the noiseprint from the authentic image to the generated one, as shown in the previous subsection. This successful transfer makes the noiseprint of the generated images not contain any traces of inconsistencies as an indication of forgery. Thus, the forgery can not be localized regardless of the authentic image used in the transfer, resulting in consistent metric values.

### Similarity with artistic style transfer approaches

Another interesting point is that the proposed approaches bear some similarities with the artistic style transfer approaches (see *Jing et al., 2020*; *Singh et al., 2021* for recent reviews on this topic). The goal of the artistic neural style transfer (NST) is to transfer a style from a famous painting style to a natural image. Similar to our proposed approaches, the goal is to transfer a noiseprint from an authentic image into a forged one. However, the proposed approaches differ from the NST in many aspects. First, the concern of the NST is to give a style to the high-level contents of an image, not the total pixels. Therefore, several NST approaches employ CNN networks designed for image classification, such as VGG network (*Simonyan & Zisserman, 2015*), to get the content representation of the image. This differs from our proposed approaches as we are concerned with all pixels to give a high-fidelity generated image. Second, traditional NST methods optimize for the output image as the proposed optimization-based approach, which makes them slow. More recent studies (*Johnson, Alahi & Fei-Fei, 2016*; *Ulyanov et al., 2016*; *Ulyanov, Vedaldi & Lempitsky, 2017*) tried to reduce the time required to generate an image with NST by training a separate CNN for each style image. Although this bears some similarity with our

noiseprint-injection-based approach, the proposed noiseprint-injection does not perform the training for each authentic image. We train the noiseprint-injector to be generic and not tailored to specific images, as discussed in 'Proposed injection based neural noiseprint transfer approach'.

### *Future directions*

Finally, in our view, the proposed framework (especially the optimization based approach) could serve as a flexible image generator that generates newly synthesized images in accordance with some constraints. Any possible combinations of the constraints can adapt the proposed framework to existing problems. Thus the strength of the proposed framework does not lie only in the successful attack of the noiseprint-based forensics, but it opens the door for other extensions for constrained image generation problems. For example, the proposed framework in this study may be adapted for attacking other noise-based forensic methods that are built on patterns such as the PRNU (*Chen et al., 2008*) and the Dark Signal Nonuniformity (DSNU) (*Berdich & Groza, 2022*). The framework could produce newly synthesized images with similar contents to forged images under the constraints of satisfying authentic patterns. Another possibility, since the proposed approaches synthesize a newly generated image, they may show robustness against generic deep-learning-based fake image detection models such as *Singh & Sharma (2021)*. However, this generalizability to other patterns or generic models has yet to be thoroughly tested and remains a subject for future research.

## CONCLUSION

This article proposes a novel generic framework for attacking the noiseprint-based forensic methods. Given forged and authentic images, the proposed framework successfully synthesizes a new image that is visually similar to the forged image but simultaneously transfers the noiseprint from the authentic image to the synthesized image to make it appear as if it is authentic. To perform this, we propose two approaches. The first is an optimization-based approach that synthesizes the generated image by minimizing the difference between its content representation with the content representation of the forged image while, at the same time, minimizing the noiseprint representation difference with the authentic one. The second approach is a noiseprint injection-based approach, which first trains a novel neural noiseprint-injector network that can inject the noiseprint of an image into another one. Then, the trained noiseprint-injector is used to inject the noiseprint from the authentic image into the forged image to produce the generated image. The effectiveness of the proposed approaches is evaluated against two common forensic tasks, the forgery localization, and camera source identification tasks. In the two tasks, the proposed approaches are able to significantly reduce several forensic accuracy scores by an average of 75% compared with the noiseprint-based forensic, while at the same time producing high fidelity images with an average PSNR of 31.5 dB and SSIM of 0.9.

### Funding

The authors received no funding for this work.

### Competing Interests

The authors declare there are no competing interests.

### Author Contributions

- Ahmed Elliethy conceived and designed the experiments, performed the experiments, analyzed the data, performed the computation work, prepared figures and/or tables, authored or reviewed drafts of the article, and approved the final draft.

### Data Availability

The source code is available at GitHub and Zenodo: https://github.com/ahmed-elliethy/nnt.

Ahmed Elliethy. (2022). ahmed-elliethy/nnt: V1.0 (V1.0). Zenodo. https://doi.org/10.5281/zenodo.7329762.

The third-party datasets are available at:

1. DSO-1 and DSI-1 Datasets: https://recodbr.wordpress.com/code-n-data/#dso1_dsi1

2. Decision-Fusion Dataset: A Framework for Decision Fusion in Image Forensics based on Dempster-Shafer Theory of Evidence, http://clem.dii.unisi.it/~vipp/datasets.html

3. Korus Dataset: https://pkorus.pl/downloads.

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
