# Peer review of "Neural noiseprint transfer: a generic noiseprint-based counter forensics framework"

_PeerJ Computer Science, doi:10.7717/peerj-cs.1359_

## Round 0.1 · original submission · Major Revisions

Dear Author,

Please improve the English language presentation of this manuscript with the help of a proficient speaker. Also, provide more details concerning the validation of the proposed approach.

Reviewer 1 ·

Basic reporting

Deep reading is required to correct some grammatical and typing errors

Experimental design

(a) Provide the algorithmic details of Figure 6.

(b) Give a detailed explanation of the different steps of Figure 6.

Validity of the findings

(a) What is the minimum execution time of the techniques proposed in Figures 5 and 6?

(b) It would also be interesting to make a comparison of the execution time with recent techniques from the literature.

(c) Is the proposed technique flexible to the resolution, type, and size of the digital images? please articulate.

Reviewer 2 ·

Basic reporting

The paper proposes a generic anti-forensics framework that is capable of fooling the Noiseprint method involving images captured by any camera device. This is a very important advantage over the existing GAN-based anti-forensics methods that need to train for each camera device that the camera model detector classifies into.
The paper proposed two deep learning-based approaches for the anti-forensics framework: 1) Optimization-based and 2) Noiseprint injection-based. Given a forged image and an authentic image, the optimization-based approach estimates an image by minimizing the difference between its content representation with the content representation of the forged image while minimizing its noiseprint representation difference from that of the authentic image. On the other hand, in the noiseprint injection-based method, a novel neural noiseprint-injector network is trained that injects the noiseprint of an image into another one. Then, the trained noiseprint-injector is used to inject the noiseprint from the authentic image into the forged one to produce the generated image. Both approaches produce very similar performance, however, the optimization-based method is 100x slower than the noiseprint-injection method. The proposed approaches are generic and do not require training for specific images or camera models.

1) The paper is well-written and easy to follow. Some minor issues are there, which are highlighted below.
2) The problem is well-formulated and the research gap and motivation are well-explained.
3) All the figures and block diagrams are of good quality and relevant to the problem.

Suggestions:
1. In the abstract, when mentioning the accuracy of the proposed method (lines 27-28), it will be better to mention the datasets on which these performances are measured.

2. In lines 42-43, the full form of PRNU should be photo-response non-uniformity instead of pixel-response non-uniformity.

3. In lines 43-44, after this line, “Additionally, there is a well-posed mathematical relation that relates an acquired image from a camera with its PRNU”, give a reference to the paper containing the relationship.

4. There is a grammatical mistake in the line, “these artifacts named as noiseprint” should be “these artifacts are named as noiseprint”

Experimental design

1. Why the L2 loss is used to penalize the differences in content and noiseprint representation? Add one sentence about the motivation for using this loss function over others.
2. How the weight-balancing factors between different loss functions are set for both methods? This should be clearly explained in the paper.
3. The experimental section should be expanded by adding more explanation about the training of both methods. For instance, in line 228, it is stated that the training image patches are extracted from two datasets, i.e., Dresden and DSO-1, for training the two approaches. However, since both method requires forged and authentic images for training, how are these image patches prepared for training? Since Dresden contains only authentic images and random crops from DSO-1 also will produce a lot of authentic patches, how are these authentic and forged cropped patches are differentiated? This should be explained clearly in the paper. Also, how many images of DSO-1 are used for training? Are the training images excluded from the test set?

Validity of the findings

1. The experimental results are promising and convincing.
2. Both the proposed methods are able to fool the Noiseprint method in both forgery localization and camera classification tasks with good accuracy.

Reviewer 3 ·

Basic reporting

In this work, the authors present two anti-forensics approaches for noiseprint-based detector. The first is optimization-based method, and the second is a neural noiseprint-injector network. In order to verify the effectivenes of the proposed approaches, they focus on two common forensics tasks: forgery localization and source camera identifcation. The paper is really hard to understand, especially in technical details. For example, it is not easy to identify the difference between optimization-based method and neural noiseprint-injector network.

Experimental design

no comment

Validity of the findings

no comment

---

## Round 0.2 · Minor Revisions

Dear Author,
Please revise and resubmit your manuscript. Thank you.

Reviewer 1 ·

Basic reporting

No comment

Experimental design

No comment

Validity of the findings

No comment

Additional comments

No comment

Reviewer 2 ·

Basic reporting

No comment

Experimental design

No comment

Validity of the findings

No comment

Annotated reviews are not available for download in order to protect the identity of reviewers who chose to remain anonymous.

Reviewer 4 ·

Basic reporting

The author proposes a counter forensic method for Noiseprint-based forgery detection method (Cozzolino, Verdoliva (2020)). I wonder how relevant it is to publish a counter forensic method that focuses on ONE forgery detection algorithm.

I would suggest including in this analysis other forgery detection methods that also make use of the noiseprint of an image:
https://arxiv.org/abs/2210.02227 (Compression fingerprints and Noiseprint)
https://arxiv.org/abs/1909.06751 (Noiseprint is added as feature along with RGB bands)

Experimental design

My main comment is regarding the experimental analysis: a comparison to other counter forensic attacks should be added. Though there are no counter forensic methods for Noiseprint specifically, I think you should compare to generic counter forensic attacks: denoising, median filter, etc.

Validity of the findings

The results are valid but I wonder how relevant they are. To assess their relevance, I think the author should test the approach on more methods that make use of the noiseprint and also compare to other counter forensic attacks.

---

## Round 0.3 · Major Revisions

Dear Author,

Please revise and resubmit your manuscript. Also, improve the English language presentation of this manuscript. Thank you.

Reviewer 1 ·

Basic reporting

I recommend the author to use passive sentences instead of active sentences.

Experimental design

No comment

Validity of the findings

(a) I recommend the author to report the results obtained (figures 3 and 4) in Section 4 and create a subsection to explain the effect of the proposed technique on the image quality.

(b) Complete the comparative analysis of tables 2 and 3 with other techniques (preferably more recent).

Additional comments

The work has been improved overall, but some details remain to be reviewed to further improve the quality of this work.

Reviewer 5 ·

Basic reporting

In this manuscript, the authors propose a novel neural noiseprint transfer framework for noiseprint-based counter forensics. Based on deep content and noiseprint representations of the forged and authentic images, the counter forensic framework is implemented through two approaches, including optimization-based and noiseprint injection-based approaches. Both approaches are evaluated on the forgery localization and camera source identification tasks, showing superior counter-forensic performance over median filtering based counter forensic method. Some comments are provided as follows:
1. Typos. “RPNU-based forensics” should be “PRNU-based forensics” in the second paragraph of Section 1.

Experimental design

2. In order to verify the effectiveness of the proposed optimization based and injection based noiseprint transfer approaches, more visualization of the extracted noiseprint should be given, including the noiseprints of the original forged image (I_f), authentic image (I_a) and generated image (I_g). To validate the noiseprint transfer performance, suggest to compare the noiseprints of I_f, I_a and I_g to demonstrate the difference between the original and the generated noiseprint.
3. The authors should clarify the training process of the optimization based approach more clearly. What are the authentic images adopted to optimize the generated image in the optimization based approach? In addition, I wonder why the authors adopt the DSO-1 dataset to train the noiseprint-injector neural network? The Dresden dataset seems to include sufficient images of different camera models for network training.

Validity of the findings

4. Due to the input images of the two proposed approaches require authentic images, I doubt that the selection of authentic image impacts the count forensic performance, which is not discussed well in the experiments. The authors should repeat the experiments several times to verify the consistency of the performance w.r.t. difference selection of authentic images.
5. Weaken the claim that “Finally, it is worth noting that the proposed approaches in this study can be adapted for attacking other noise-based forensic methods that patterns such as the PRNU” since there are not sufficient experimental results to support it.

Annotated reviews are not available for download in order to protect the identity of reviewers who chose to remain anonymous.

·

Basic reporting

The paper is clear and unambiguous.
- I feel the Abstract can be improved a little so that the reader understands the meaning of the noiseprint and how this proposed system works in simple sentences. The Introduction section explains it clearly and in simple language, a similar approach can be taken for the Abstract.

- There are a few spelling mistakes like PRNU is written as RPNU in line 50

- In Figure 5 and Figure 6 'BN' is batch normalization? It's not clear in the explanation

Experimental design

- In section 4.2 if you can, please add, examples of how localization of the fake images failed after injection of noiseprint. The lower accuracy results are fine, but to bring more explainability a few examples of that even localization detection failed after the proposed model noiseprint injection can be more helpful

- Appreciate that besides the F1 score, MCC was also calculated, good work

Validity of the findings

- Image forgery can be done in multiple ways - Image Splicing, Copy/Move and resampling
As far as I understood, this experiment to validate the proposed model had more image splicing and cropping images. Will this model be equally good for copy/move, resampling, and multiple compression manipulations?

- The image was fake but with a similar noiseprint which could not be detected.
Did we try to test the generated images using another generic deep-learning fake image detection model? If we can do that then the proposed model's robustness is very good. Any deep-learning models like the one described below can be used.
Singh, B. and Sharma, D,K. SiteForge: Detecting and localizing forged images on microblogging platforms using deep convolutional neural network. Computers & Industrial Engineering, vol. 162, (Dec 2021). https://doi.org/10.1016/j.cie.2021.107733

Additional comments

The paper is very well written. I appreciate the author for his proposed model and its novelty. The model can create fake images with similar noiseprint is really good. The model is generic and experimented on different cameras proving its robustness.
There are a few minor review comments and questions (as provided in the sections above) which can be added by the author to make the work more authentic and prove its robustness.

Another request I have with the author is the utility of this model. Besides generating a good undetectable fake images dataset what other practical utility can this proposed model be used for? It would be great if more utilization examples are provided by the author.

---

## Round 0.4 · accepted · Accept

Author has addressed all of the reviewers' comments.

Reviewer 1 ·

Basic reporting

No comment.

Experimental design

No comment.

Validity of the findings

No comment.

Additional comments

No comment.

Reviewer 5 ·

Basic reporting

Thank the authors for their efforts. The authors have made a good response to my concerns. This article is suggested to be accepted as it is.

Experimental design

no comment

Validity of the findings

no comment

Additional comments

no comment

·

Basic reporting

The research article is more clear now. The review comments are incorporated and well explained

Experimental design

More information and image are added as requested which proves the reliability and robustness of the proposed model

Validity of the findings

The findings and tables are valid and capture enough information for the comparison and validity

Additional comments

The article is in good shape now and has plugged in a few gaps.